# WEAK NAS PREDICTOR IS ALL YOU NEED

## ABSTRACT

Neural Architecture Search (NAS) finds the best network architecture by exploring the architecture-to-performance manifold. It often trains and evaluates a large amount of architectures, causing tremendous computation cost. Recent predictor-based NAS approaches attempt to solve this problem with two key steps: sampling some architecture-performance pairs and fitting a proxy accuracy predictor. Existing predictors attempt to model the performance distribution over the whole architecture space, which could be too challenging given limited samples. Instead, we envision that this ambitious goal may not be necessary if the final aim is to find the best architecture. We present a novel framework to estimate weak predictors progressively. Rather than expecting a single strong predictor to model the whole space, we seek a progressive line of weak predictors that can connect a path to the best architecture, thus greatly simplifying the learning task of each predictor. It is based on the key property of the predictors that their probabilities of sampling better architectures will keep increasing. We thus only sample a few well-performed architectures guided by the predictive model, to estimate another better weak predictor. By this coarse-to-fine iteration, the ranking of sampling space is refined gradually, which helps find the optimal architectures eventually. Experiments demonstrate that our method costs fewer samples to find the top-performance architectures on NAS-Benchmark-101 and NAS-Benchmark-201, and it achieves the state-of-the-art ImageNet performance on the NASNet search space.

## 1 INTRODUCTION

Neural Architecture Search (NAS) has become a central topic in recent years with great progress (Liu et al., 2018b; Luo et al., 2018; Wu et al., 2019; Howard et al., 2019; Ning et al., 2020; Wei et al., 2020; Luo et al., 2018; Wen et al., 2019; Chau et al., 2020; Luo et al., 2020). Methodologically, all existing NAS methods try to find the best network architecture by exploring the architecture-to-performance manifold, such as reinforced-learning-based (Zoph & Le, 2016), evolution-based (Real et al., 2019) or gradient-based Liu et al. (2018b) approaches. In order to cover the whole space, they often train and evaluate a large amount of architectures, thus causing tremendous computation cost.

Recently, predictor-based NAS methods alleviate this problem with two key steps: one sampling step to sample some architecture-performance pairs, and another performance modeling step to fit the performance distribution by training a proxy accuracy predictor. An in-depth analysis of existing methods (Luo et al., 2018) founds that most of those methods (Ning et al., 2020; Wei et al., 2020; Luo et al., 2018; Wen et al., 2019; Chau et al., 2020; Luo et al., 2020) attempt to model the performance distribution over the whole architecture space. However, since the architecture space is often exponentially large and highly non-convex, modeling the whole space is very challenging especially given limited samples. Meanwhile, different types of predictors in these methods have to demand handcraft design of the architecture representations to improve the performance.

In this paper, we envision that the ambitious goal of modeling the whole space may not be necessary if the final goal is to find the best architecture. Intuitively, we assume the whole space could be divided into different sub-spaces, some of which are relatively good while some are relatively bad. We tend to choose the good ones while neglecting the bad ones, which makes sure more samples will be used to model the good subspace precisely and then find the best architecture. From another perspective, instead of optimizing the predictor by sampling the whole space as well as existing methods, we propose to jointly optimize the sampling strategy and the predictor learning, which helps achieve better sample efficiency and prediction accuracy simultaneously.

Based on the above motivation, we present a novel framework that estimates a series of weak predictors progressively. Rather than expecting a strong predictor to model the whole space, we instead seek a progressive evolving of weak predictors that can connect a path to the best architecture. In this way, it greatly simplifies the learning task of each predictor. To ensure moving the best architecture along the path, we increase the sampling probability of better architectures guided by the weak predictor at each iteration. Then, the consecutive weak predictor with better samples will be trained in the next iteration. We iterate until we arrive at an embedding subspace where the best architectures reside. The weak predictor achieved at the final iteration becomes the dedicated predictor focusing on such a fine subspace and the best performed architecture can be easily predicted.

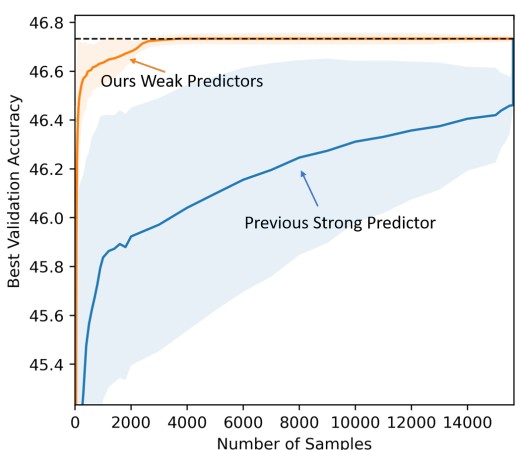

Figure 1: Comparison between iterative weak predictors and non-iterative strong predictor on NAS-Bench-201 ImageNet subset. Our method significantly reduces the needed amount of samples to reach the optimal architecture.

Compared to existing predictor-based NAS, our method has several merits. First, since only weak predictors are required to locate the good subspace, it yields better sample efficiency. On NAS-Benchmark-101 and NAS-Benchmark-201, it costs significantly fewer samples to find the top-performance architecture than existing predictor-based NAS methods. Second, it is much less sensitive to the architecture representation (e.g., different architecture embeddings) and the predictor formulation design (e.g., MLP, Gradient Boosting Regression Tree, Random Forest). Experiments show our superior robustness in all their combinations. Third, it is generalized to other search spaces. Given a limited sample budget, it achieves the state-of-the-art ImageNet performance on the NASNet search space.

## 2 OUR APPROACH

### 2.1 REVISIT PREDICTOR-BASED NEURAL ARCHITECTURE SEARCH

Neural Architecture Search (NAS) finds the best network architecture by exploring the architecture-to-performance manifold. It can be formulated as an optimization problem. Given a search space of network architectures $X$ and a discrete architecture-to-performance mapping function $f : X \to P$ from architecture set $X$ to performance set $P$, the objective is to find the best neural architecture $x^*$ with the highest performance $f(x)$ in the search space $X$:

$$x^* = \arg\max_{x \in X} f(x) \tag{1}$$

A naïve solution is to estimate the performance mapping $f(x)$ through the full search space, however, it is prohibitively expensive since all architectures have to be exhaustively trained from scratch. To address this problem, predictor-based NAS learns a proxy predictor $\tilde{f}(x)$ to approximate $f(x)$ using some architecture-performance pairs , which significantly reduces the training cost. In general, predictor-based NAS can be formulated as:

$$x^* = \arg\max_{x \in X} \tilde{f}(x|S)$$
$$\text{s.t. } \tilde{f} = \arg\min_{S, \tilde{f} \in \tilde{\mathcal{F}}} \sum_{s \in S} \mathcal{L}(\tilde{f}(s), f(s)) \tag{2}$$

where $\mathcal{L}$ is the loss function for the predictor $\tilde{f}$, $\tilde{\mathcal{F}}$ is a set of all possible approximation to $f$, $S := \{S \subseteq X \mid |S| \leq C\}$ is the training pairs for predictor $\tilde{f}$ given sample budget $C$. Here, $C$ is directly correlated to the total training cost. Our objective is to minimize the loss $\mathcal{L}$ of the predictor $\tilde{f}$ based on some sampled architectures $S$.

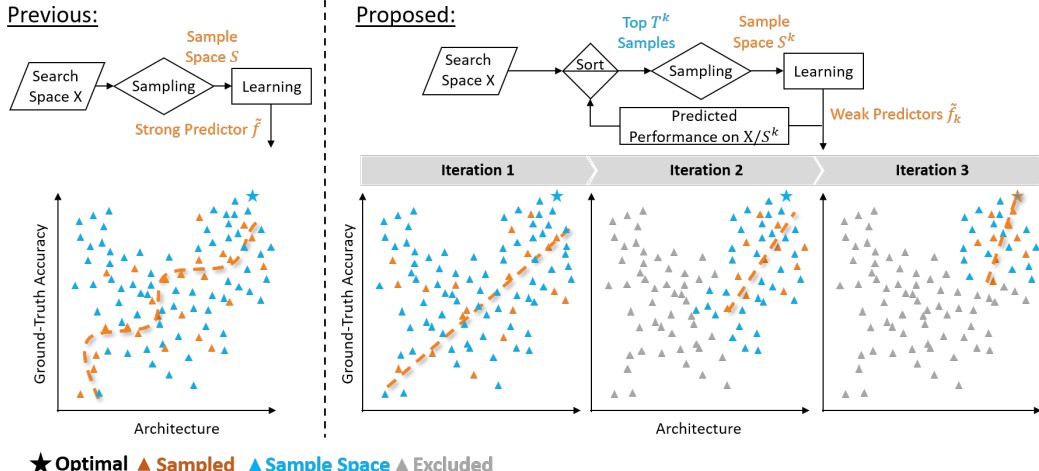

Figure 2: An illustration of our progressive weak predictors approximation (best viewed in color). Previous predictor-based NAS uniformly sampled in the whole search space to fit a strong predictor; Ours progressively shrink the sample space smaller based on predictions from previous weak predictors and train new weak predictor towards subspace of better architectures

Previous predictor-based NAS methods attempt to solve Equation 2 with two key steps: (1) *sampling* some architecture-performance pairs and (2) *learning* a proxy accuracy predictor. First, a common practice in previous work is to sample training pairs $S$ uniformly from the search space $X$ to learn the predictor. Such a sampling is inefficient considering that the goal of NAS is to find a subspace of well-performed architectures in the search space. A biased sampling strategy towards the well-performed architectures can be more desirable. Second, given such pairs $S$, previous predictor-based NAS uses a predictor $\tilde{f}$ to model the performance distribution over the whole architecture space. Since the architecture space is often enormously large and highly non-convex, it is too challenging to model the whole space given the limited samples.

## 2.2 PROGRESSIVE WEAK PREDICTORS APPROXIMATION

We envision that the above ambitious goal may not be necessary if the final aim of NAS is to find the best architecture. We argue that sampling $S$ and learning $\tilde{f}$ should be co-evolving instead of a one-time deal as done in existing predictor-based NAS. Demonstrated in Figure 2, rather than expecting a single strong predictor to model the whole space at one time, we progressively evolve our weak predictors to sample towards subspace of best architectures, thus greatly simplifying the learning task of each predictor. With these coarse-to-fine iterations, the ranking of sampling space is refined gradually, which helps find the optimal architectures eventually.

Thus, we propose a novel coordinate descent way to *jointly* optimize the *sampling* and *learning* stages in predictor-based NAS progressively, which can be formulated as following:

$$\text{Sampling Stage:} \quad \tilde{P}^k = \{\tilde{f}_k(s) | s \in X \setminus S^k\} \tag{3}$$

$$S^{k+1} = \underset{T^k}{\arg\max}(\tilde{P}^k) \cup S^k \tag{4}$$

$$\text{Learning Stage:} \quad x^* = \underset{x \in X}{\arg\max} \, \tilde{f}(x | S^{k+1})$$

$$\text{s.t.} \, \tilde{f}_{k+1} = \underset{\tilde{f}_k \in \tilde{\mathcal{F}}}{\arg\min} \sum_{s \in S^{k+1}} \mathcal{L}(\tilde{f}(s), f(s)) \tag{5}$$

Suppose our iterative methods has $K$ iterations, at $k$-th iteration where $k = 1, 2, \ldots K$, we initialize our training set $S^1$ by randomly sampling a few samples from $X$ to train an initial predictor $\tilde{f}_1$. We then jointly optimize the sampling set $S^k$ and predictor $\tilde{f}_k$ in a progressive manner for $K$ iterations.

**Sampling Stage** We first sort all the architectures in the search space $X$ according to its predicted performance $\tilde{P}^k$ at every iteration $k$. Given the sample budget, we then sample new architectures $S^{k+1}$ among the top $T^k$ ranked architectures.

**Learning Stage** We learn a predictor $\tilde{f}^k$, where we want to minimize the the loss $\mathcal{L}$ of the predictor $\tilde{f}^k$ based on sampled architectures $S^k$. We then evaluate all the architectures $X$ in the search space using the learned predictor $\tilde{f}^k$ to get the predicted performance $\tilde{P}^k$.

**Progressive Approximation** Through the above alternative iteration, the predictor $\tilde{f}^k$ would guide the sampling process to gradually zoom into the promising architecture samples. In addition, the good performing samples $S^{k+1}$ sampled from the promising architecture samples would in term improve the performance of the predictor $\tilde{f}^{k+1}$ in the well-performed architectures.

To demonstrate the effectiveness of our iterative scheme, Figure 3 (a) shows the progressive procedure of finding the optimal architecture $x^*$ and learning the predicted best architecture $\tilde{x}_k^*$ over 5 iterations. As we can see, the optimal architecture and the predicted best one are moving towards each other closer and closer, which indicates that the performance of predictor over the optimal architecture(s) is growing better. In Figure 3 (b), we use the error *empirical distribution function* (EDF) proposed in (Radosavovic et al., 2020) to visualize the performance distribution of architectures in the subspace. We plot the EDF of the top-200 models based on the predicted performance over 5 iterations. As shown in Figure 3 (b), the subspace of top-performed architectures is consistently evolving towards more promising architecture samples over 5 iterations. In conclusion, the probabilities of sampling better architectures through these progressively improved weak predictors indeed keep increasing, as we desire them to.

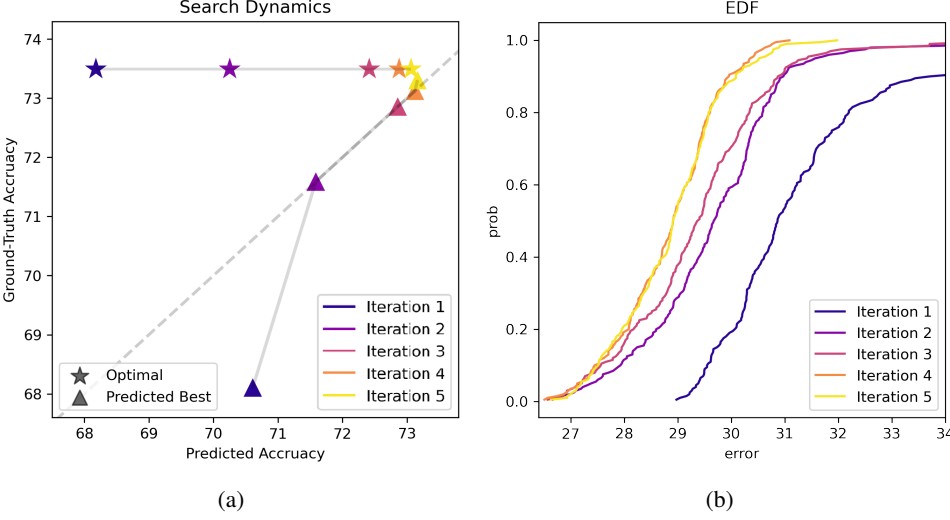

(a)                                           (b)

Figure 3: Visualization of the search dynamics. (a): The trajectory of Predicted Best architecture and Global Optimal through out 5 iterations; (b): Error *empirical distribution function* (EDF) of predicted Top 200 architectures through out 5 iterations

## 2.3 GENERALIZABILITY ON PREDICTORS AND FEATURES

Here we analyze the generalizability of our method and demonstrate its robustness on different predictors and features. In predictor-based NAS, the objective of learning the predictor $\tilde{f}$ can be formulated as a regression problem (Wen et al., 2019) or a ranking (Ning et al., 2020) problem. The choice of predictors is diverse, and usually critical to final performance (e.g. MLP (Ning et al., 2020; Wei et al., 2020), LSTM (Luo et al., 2018), GCN (Wen et al., 2019; Chau et al., 2020), Gradient Boosting Tree (Luo et al., 2020)). To illustrate our framework is generalizable and robust to the specific choice of predictors, we compare the following predictor variants.

- Multilayer perceptron (MLP): MLP is the baseline commonly used in predictor-based NAS (Ning et al., 2020) due to its simplicity. Here we use a 4-layer MLP with hidden layer dimension of (1000, 1000, 1000, 1000) which is sufficient to model the architecture encoding.

- Gradient Boosting Regression Tree (GBRT): Tree-based methods have recently been preferred in predictor-based NAS (Luo et al., 2020; Siems et al., 2020) since it is more suitable to model discrete representation of the architectures. Here, we use the Gradient Boosting Regression Tree based on XGBoost (Chen & Guestrin, 2016) implementation.

- Random Forest: Random Forrest is another variant of tree-based predictor, it differs from Gradient Boosting Trees in that it combines decisions at the end instead of along each hierarchy, and thus more robust to noise.

The selection of features to represent the architecture search space and learn the predictor is also sensitive to the performance. Previous methods tended to hand craft the feature for the best performance (e.g., raw architecture encoding (Wei et al., 2020), supernet statistic (Hu et al., 2020)). To demonstrate our framework is robust across different features, we compare the following features.

- One-hot Vector: In NAS-Bench-201(Dong & Yang, 2020), its DART style search space fixed the graph connectivity, so one-hot vector is used to encode the choice of operator.

- Adjacency Matrix: In NAS-Bench-101, we used the encoding scheme as well as (Ying et al., 2019; Wei et al., 2020), where a $7 \times 7$ adjacency matrix represents the graph connectivity and a 7-dimensional vector represents the choice of operator, on every node.

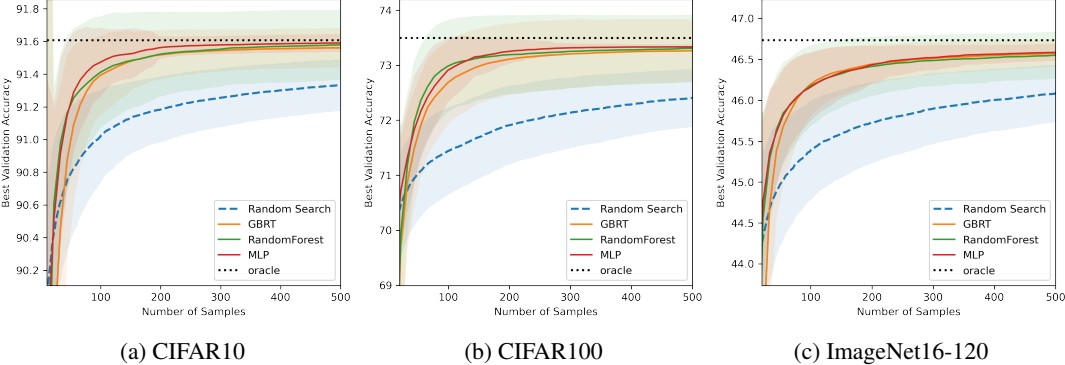

(a) CIFAR10      (b) CIFAR100      (c) ImageNet16-120

Figure 4: Evaluations of robustness across different predictors on NAS-Bench-201

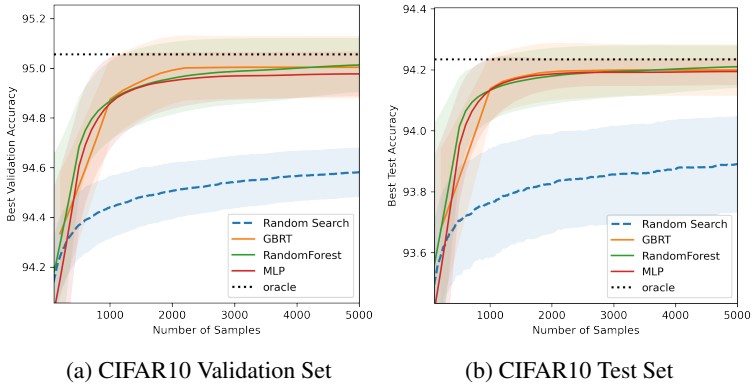

(a) CIFAR10 Validation Set      (b) CIFAR10 Test Set

Figure 5: Evaluations of robustness across different predictors on NAS-Bench-101

We compare the robustness across different predictors under our framework shown in Figure 4. We can see that all predictors perform similarly among different target datasets. As shown in Figure 4 with Figure 5, although different architecture encoding methods are used, our method can perform similarly well among different predictors, which demonstrates that our proposed method is robust to different predictors and features selection.

## 3 EXPERIMENTS

### 3.1 SETUP

**NAS-Bench-101** (Ying et al., 2019) is one of the first datasets used to benchmark NAS algorithms. The dataset provides a Directed Acyclic Graph (DAG) based cell structure, while (1) The connectivity of DAG can be arbitrary with a maximum number of 7 nodes and 9 edges (2) Each nodes on the DAG can choose from operator of $1 \times 1$ convolution, $3 \times 3$ convolution or $3 \times 3$ max-pooling. After removing duplications, the dataset consists of 423,624 diverse architectures trained on CIFAR10 dataset with each architecture trained for 3 trials.

**NAS-Bench-201** (Dong & Yang, 2020) is another recent NAS benchmark with a reduced DARTS-like search space. The DAG of each cell is fixed similar to DARTS(Liu et al., 2018b), however we can choose from 5 different operations ($1 \times 1$ convolution, $3 \times 3$ convolution, $3 \times 3$ avg-pooling, skip, no connection) on each of the 6 edges totaling a search space of 15,625 architectures. The dataset is trained on 3 different datasets (CIFAR10/CIFAR100/ImageNet16-120) with each architecture trained for 3 trials.

For experiments on both benchmarks, we followed the same setting as (Wen et al., 2019). We use the validation accuracy as search signal, while test accuracy is only used for reporting the accuracy on the model that was selected at the end of a search. Since the best performing architecture on the validation and testing set does not necessarily match, we also report the performance on finding the oracle on the validation set of our NAS algorithm in the following experiments.

**Open Domain Search:** we follow the same NASNet search space used in (Zoph et al., 2018) to directly search for best-performing architectures on ImageNet. Due to the huge computational costs needed to train and evaluate architecture performance on ImageNet, we leverage a weight-sharing supernet approach (Guo et al., 2019) and use supernet accuracy as a performance proxy indicator.

### 3.2 COMPARISON TO STATE-OF-THE-ART (SOTA) METHODS

We evaluate our method on both NAS-Bench-101 and NAS-Bench-201. We also apply our method to open domain search directly on ImageNet dataset using NASNet search space.

**NAS-Bench-101**

We conduct experiments on the popular NAS-Bench-101 benchmark and compare with multiple popular methods (Real et al., 2019; Wang et al., 2019b;a; Luo et al., 2018; Wen et al., 2019). We first study the performance by limiting the number of queries. In Table 1, we vary the number of queries used in our method by changing the number of iterations. It is clear to see that, the searched performance consistently improves as more iterations are used. When compared to the results from popular predictor-based NAS methods, such as NAO (Luo et al., 2018) and Neural Predictor (Wen et al., 2019), our method (a) reaches higher search performance provided with the same query budget for training; and (b) uses fewer samples towards the same accuracy goal.

We then plot the best accuracy against number of samples in Figure 6 to show the sample efficiency on both validation and test set of NAS-Bench-101, we can see that our method consistently requires fewer sample to reach higher accuracy, compared to Random Search and Regularized Evolution.

Finally, Table 2 shows that our method significantly outperforms baselines in terms of sample efficiency. Specifically, our method costs $44 \times$, $20 \times$, $17 \times$, and $2.66 \times$ less samples to reach the optimal architecture, compared to Random Search, Regularized Evolution (Real et al., 2019), MCTS (Wang et al., 2019b), LaNAS (Wang et al., 2019a), respectively.

**NAS-Bench-201**

We further evaluate our method on NAS-Bench-201. Since it is relatively newly released, we compare with two baseline methods Regularized Evolution (Real et al., 2019) and random search using our own implementation. Shown in Table 3, we conduct searches on all three subsets (CIFAR10, CIFAR100, ImageNet16-120) and report the average number of samples needed to reach global optimal over 250 runs. It is obvious to see that our method requires the minimum number of samples among all settings.

Table 1: Comparsion to SOTA on NAS-Bench-101 by limiting the total amount of queries

| Method | #Queries | Test Acc.(%) | SD(%) | Test Regret(%) |
|---|---|---|---|---|
| Random Search | 2000 | 93.64 | 0.25 | 0.68 |
| NAO (Luo et al., 2018) | 2000 | 93.90 | 0.03 | 0.42 |
| Reg Evolution (Real et al., 2019) | 2000 | 93.96 | 0.05 | 0.36 |
| Neural Predictor (Wen et al., 2019) | 2000 | 94.04 | 0.05 | 0.28 |
| Ours (Non-iterative) | 2000 | 93.92 | 0.082 | 0.04 |
| Ours (1 iteration) | 100 | 93.47 | 0.183 | 0.85 |
| Ours (4 iterations) | 400 | 94.15 | 0.146 | 0.17 |
| Ours (10 iterations) | 1000 | 94.20 | 0.077 | 0.12 |
| Ours (20 iterations) | **2000** | **94.22** | **0.028** | **0.10** |

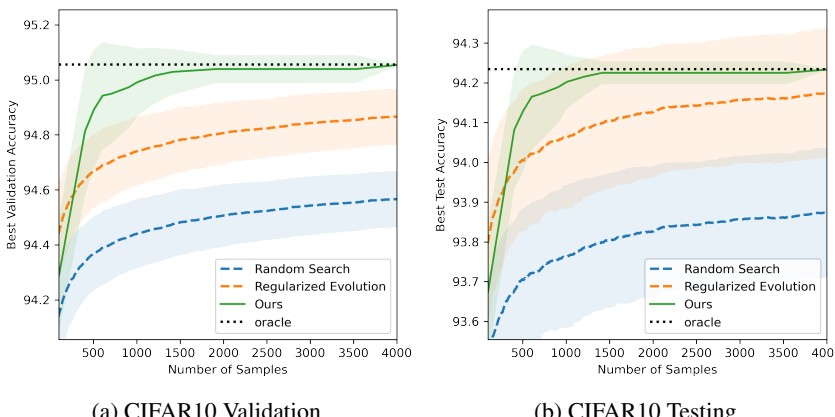

(a) CIFAR10 Validation        (b) CIFAR10 Testing

Figure 6: Comparison to SOTA on NAS-Bench-101 by varying number of samples. Central lines demonstrate the average, and the shade regions depict the confidence intervals

Table 2: Comparison on the number of samples required to find the global optimal architecture over 100 runs on NAS-Bench-101

| Method | #Queries |
|---|---|
| Random Search | 188139.8 |
| Reg Evolution (Real et al., 2019) | 87402.7 |
| MCTS (Wang et al., 2019b) | 73977.2 |
| LaNAS (Wang et al., 2019a) | 11390.7 |
| Ours | **4275.3** |

We also conduct a controlled experiment by varing the number of samples. As in Figure 7, our average performance over different number of samples yields a clear gain over Regularized Evolution (Real et al., 2019) in all three subsets. Our confidence interval is also tighter than Regularized Evolution, showcasing our method's superior stability/reliability.

**Open Domain Search**

In order to demonstrate our method's generalizability, we further apply it to open domain search without ground-truth. We adopt the popular search space from NASNet and compare with several popular methods (Zoph et al., 2018; Real et al., 2019; Liu et al., 2018a; Luo et al., 2018) with the utilization of number of samples reported. Shown in Table 5, it is clear to see that, using fewest samples among all, our method achieves state-of-the-art ImageNet top-1 error with similar number of parameters and FLOPs. Our searched architecture is also competitive to expert-design networks. Comparing with the previous SOTA predictor-based NAS method (Luo et al., 2018), our method

Table 3: Comparison on the number of samples required to find the global optimal over 250 runs on NAS-Bench-201

| Method | CIFAR10 | | CIFAR100 | | ImageNet16-120 | |
|---|---|---|---|---|---|---|
| | valid | test | valid | test | valid | test |
| Random Search | 7873.98 | 7782.12 | 7716.85 | 7621.24 | 7776.55 | 7726.15 |
| Reg Evolution (Real et al., 2019) | 478.41 | 563.25 | 399.78 | 438.24 | 802.57 | 715.12 |
| Ours | **119.17** | **186.30** | **86.08** | **89.26** | **159.33** | **176.61** |

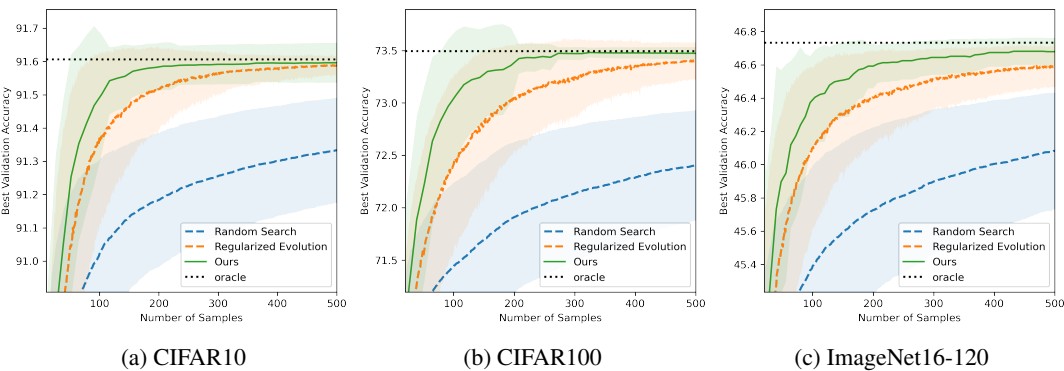

(a) CIFAR10       (b) CIFAR100       (c) ImageNet16-120

Figure 7: Comparison to SOTA on NAS-Bench-201 by varying number of samples. Central line demonstrates the average, shade demonstrates the confidence interval

reduces 0.9% top-1 error, using the same number of samples, which is significant. This experiment well proves that our method is robust, generalizable, and can be effectively applied to real-world open domain search.

Table 4: Compared to SOTA results on ImageNet using NASNet search space

| Method | #Queries | Top-1 Err.(%) | Top-5 Err.(%) | Params(M) | FLOPs(M) |
|---|---|---|---|---|---|
| MobileNetV2 | Manual | 25.3 | - | 6.9 | 585 |
| ShuffletNetV2 | Manual | 25.1 | - | 5.0 | 591 |
| EfficientNet-B0 | Manual | 23.7 | 6.8 | 5.3 | 390 |
| NASNet-A (Zoph et al., 2018) | 20000 | 26.0 | 8.4 | 5.3 | 564 |
| AmoebaNet-A (Real et al., 2019) | 10000 | 25.5 | 8.0 | 5.1 | 555 |
| PNAS (Liu et al., 2018a) | 1160 | 25.8 | 8.1 | 5.1 | 588 |
| NAO (Luo et al., 2018) | 1000 | 24.5 | 7.8 | 6.5 | 590 |
| Ours | 1000 | **23.6** | **6.8** | **5.9** | **597** |

## 4   CONCLUSION

In this paper, we present a novel predictor-based NAS framework that progressively shrinks the sampling space, by learning a series of weak predictors that can connect towards the best architectures. We argue that using a single strong predictor to model the whole search space with limited samples may be too challenging a task and seemingly unnecessary. Instead by co-evolving the sampling stage and learning stage, our weak predictors can progressively evolve to sample towards the subspace of best architectures, thus greatly simplifying the learning task of each predictor. Extensive experiments on popular NAS benchmarks prove that proposed method is sample-efficient and robust to various combinations of predictors and architecture encoding means. We further apply our method to open domain search and demonstrate its generalization. Our future work will investigate how to jointly the predictor and encoding in our framework.

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

## A MORE COMPARSION ON NASNET SEARCH SPACE

In Table 5, we show more comparsion to representative Gradient-based methods including SNAS(Xie et al., 2018), DARTS(Liu et al., 2018b), P-DARTS(Chen et al., 2019), PC-DARTS(Xu et al., 2019), DS-NAS(Xu et al., 2019).

Table 5: More Comparsion to SOTA results on ImageNet using NASNet search space. WS denote Weight-Sharing

| Model | Methods | #Queries | Top-1 Err.(%) | Top-5 Err.(%) | Params(M) | FLOPs(M) | GPU Days |
|---|---|---|---|---|---|---|---|
| MobileNetV2 | | - | 25.3 | - | 6.9 | 585 | - |
| ShuffletNetV2 | Manual | - | 25.1 | - | 5.0 | 591 | - |
| EfficientNet-B0 | | - | 23.7 | 6.8 | 5.3 | 390 | - |
| SNAS(Xie et al., 2018) | | - | 27.3 | 9.2 | 4.3 | 522 | 1.5 |
| DARTS(Liu et al., 2018b) | | - | 26.9 | 9.0 | 4.9 | 595 | 4.0 |
| P-DARTS(Chen et al., 2019) | Gradient-based | - | 24.4 | 7.4 | 4.9 | 557 | 0.3 |
| PC-DARTS(Xu et al., 2019) | | - | 24.2 | 7.3 | 5.3 | 597 | 3.8 |
| DS-NAS(Xu et al., 2019) | | - | 24.2 | 7.3 | 5.3 | 597 | 10.4 |
| NASNet-A (Zoph et al., 2018) | | 20000 | 26.0 | 8.4 | 5.3 | 564 | 2000 |
| AmoebaNet-A (Real et al., 2019) | Sample-based | 10000 | 25.5 | 8.0 | 5.1 | 555 | 3150 |
| PNAS (Liu et al., 2018a) | | 1160 | 25.8 | 8.1 | 5.1 | 588 | 200 |
| NAO (Luo et al., 2018) | | 1000 | 24.5 | 7.8 | 6.5 | 590 | 200 |
| Ours | Sample-based + WS | 1000 | **23.6** | **6.8** | **5.9** | **597** | 1.08 |

## B VISUALIZATION OF NASBENCH SEARCH SPACE

Figure 8 illustrates the histogram of architecture performance in NASBench-101/201, we also accompany a zoomed in view of the histogram, those histograms clearly show that the NAS search spaces bias heavily towards good performing architectures.

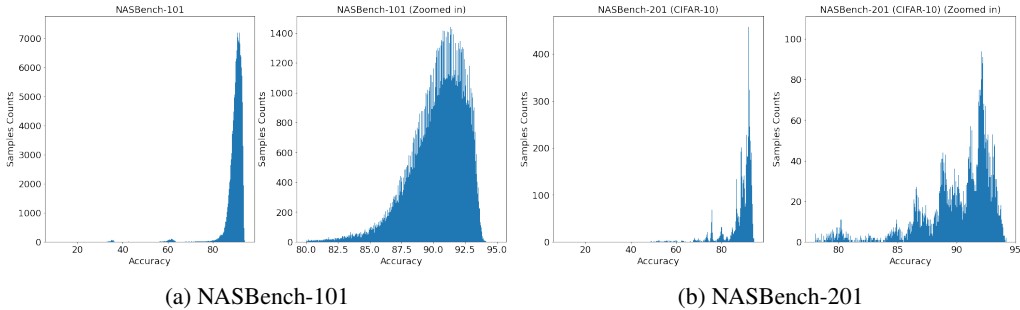

(a) NASBench-101           (b) NASBench-201

Figure 8: (a) Histogram of Architecture Performance in NASBench-101 (b) Histogram of Architecture Performance in NASBench-201 (CIFAR10)

