# OpenReview forum: "Weak NAS Predictor Is All You Need"
_ICLR.cc/2021/Conference — Reject_

### Official Review · AnonReviewer3 · 2020-10-24
**This work is more like an incremental improvement to the neural predictor. It gets a small performance improvement by learning in a better subspace.**

**Rating:** 4
**Confidence:** 4

**Review:**

Pros:
- A progressive method for neural predictor-based NAS is proposed, and it shows better performance than previous methods.
- Experiments on the NAS-bench-101 and NAS-bench-201, as well as on the NASNet search space to validate the effectiveness of the method.
- The proposed method is conceptually simple and efficient.

Cons:
- This work is more like an incremental improvement to the neural predictor. It gets a small performance improvement by learning in a maybe better subspace.
- Is it effective to simply choose TopK models? As the author said, the learning space is not convex thus TopK samples may contain many local-optima and make the learning of predictor difficult even failure (all the predicted Accs fall within a small range).
- In Fig. 1, why is there a sudden increase for the strong predictor (at near 15625 samples.)
- What if the search space is huge, like 10^20 in many SOTA NAS search space, rather than the small models in NAS-Benchs, the proposed method seems inefficient in this case. Will it still work?
- What's the difference and advantages between the proposed model, and choosing the top 10% of the models and training a single predictor?

---

> ### Author Response · Authors · 2020-11-23
> **Response to Reviewer3 (part2/2)**
>
> *Q5: What's the difference and advantages between the proposed model, and choosing the top 10\% of the models and training a single predictor?*
>
> We followed the setting and conducted the experiment in Table 3. We randomly sampled the Top 10%/20%/50% or all sampled models to train a single predictor, and we found all of these ways performed worse than our method. It demonstrates that training a single predictor is difficult and sub-optimal because the NAS search space is very complex. It also evidences our key idea of using a series of weak predictors is not trivial at all.
>
> Table 3: Comparison to the simple baselines by training a single predictor with TopX% sample on NAS-Bench-101 over 50 runs
>
> | Method   |      \#Queries      |  Test Acc.(\%) |  Std(\%) |  Test Regret(\%) |
> |----------|:-------------:|------:|------:|------:|
> | Random Search | 2000 | 93.64 | 0.25 | 0.68
> | NAO  | 2000 | 93.90 | 0.03 | 0.42
> | Reg Evolution  | 2000 | 93.96 | 0.05 | 0.36
> | Neural Predictor  | 2000 | 94.04 | 0.05 | 0.28
> | Ours (Single Predictor) (Top 10%) | 2000 | 93.65 | 0.025 | 0.67
> | Ours (Single Predictor) (Top 20%) | 2000 | 93.67 | 0.031 | 0.65
> | Ours (Single Predictor) (Top 50%) | 2000 | 93.67 | 0.023 | 0.65
> | Ours (Single Predictor) (All) | 2000 | 93.92 | 0.082 | 0.04
> | Ours (20 Predictors) | 2000 | 94.22 | 0.028 | 0.10

---

> ### Author Response · Authors · 2020-11-23
> **Response to Reviewer3 (part1/2)**
>
> We appreciate your time and efforts in reviewing our paper!
>
> *Q1: Incremental improvement to the neural predictor. It gets a small performance improvement by learning in a maybe better subspace.*
>
> Our contribution in nontrivial compared with existing neural predictors since these tend to model the whole architecture space. In contrast, our key insight is that "We not do need to model the whole architecture space, if the goal of NAS is finding the best architecture(s) in the search space", which then motivated the proposed weak predictors. We believe that it is **important** to share this observation to the community.
> Based on this key observation, we are the first to propose to progressively learn a series of weak predictors to efficiently sample a path to the best architectures. As shown in Figure 3 of the main paper, the probability of sampling better architectures keeps increasing, thus in each iteration we only need to sample a few architectures, which largely improves the efficiency.
>
> In the experiment, we also demonstrate that it is crucial for predictor-based NAS to learn better subspace. As shown in Table 1, to achieve the same Test Accuracy to existing predictor based NAS SOTA methods (e.g., Neural Predictor), the samples required in our method can be ~5.71X less.
>
> Table 1: Comparison to SOTA on NAS-Bench-101 by limiting the total amount of queries
>
> | Method   |      \#Queries      |  Test Acc.(\%) |  Std(\%) |  Test Regret(\%) |
> |----------|:-------------:|------:|------:|------:|
> | Random Search | 2000 | 93.64 | 0.25 | 0.68
> | NAO  | 2000 | 93.90 | 0.03 | 0.42
> | Reg Evolution  | 2000 | 93.96 | 0.05 | 0.36
> | Neural Predictor  | **2000** | **94.04** | **0.05** | **0.28**
> | Ours | **350** | **94.04** | **0.13** | **0.28**
>
> *Q2: Is it effective to simply choose TopK models? As the author said, the learning space is not convex thus TopK samples may contain many local-optima and make the learning of predictor difficult even failure (all the predicted Accs fall within a small range).*
>
> This is a really insightful question. In our method, we sample a fix amount (10/100) of models in Top-K (K=100/1000) models instead of choosing all of Top K models to alleviate the effect of noisy (local-optima) samples. In the following Table 2, We did an ablation study to know whether the sampling is indispensable. We set the budget to 2000 Queries. For the experiment without sampling, we sample 100 samples out of Top 100/1000 models for 20 iterations. For the experiment with sampling, we sample 100 sample out of Top 1000 models for 20 iterations. We find that the latter (w/ sampling) achieves slightly better performance with lower variance between runs than the former (w/o sampling).
>
> Table 2: Comparison to without/with sampling in TopK models on NAS-Bench-101 over 50 runs.
>
> | Method   |      \#Queries      |  Test Acc.(\%) |  Std(\%) |  Test Regret(\%) |
> |:---------|:-------------:|------:|------:|------:|
> | Ours (Top 100, w/o sampling) | 2000 | 94.1739 | 0.038 | 0.1461 |
> | Ours (Sample 100 out of Top 1000, w/ sampling) | 2000 | 94.2203 | 0.028 | 0.0997 |
>
> *Q3: In Fig. 1, why is there a sudden increase for the strong predictor (at near 15625 samples)*
>
> In Figure 1 of our paper, the sudden increase at the end indicates that there is a gap for strong predictor to find the optimal architecture, since a single strong predictor cannot extrapolate well on unseen well-performing architectures. This motivates us to use a progressive line of weak predictors to bridge this gap, which helps us connect a path to the best architecture.
>
> *Q4: What if the search space is huge, like 10^20 in many SOTA NAS search space, rather than the small models in NAS-Benchs, the proposed method seems inefficient in this case. Will it still work?*
>
> Yes, our method works well for a larger search space. Table 4 of our paper shows our result on NASNet ImageNet search space with 10^25 architectures. Instead of evaluating the performance of all 10^25 architectures at every iteration, we only need to sample 1000 models per iteration for evaluation and re-ranking. Based on this setting, we can achieve SOTA NASNet search space performance of 76.4\% with 597 MFLOPs budget without bells and whistles.

---

### Official Review · AnonReviewer2 · 2020-10-26
**An interesting and inspirational paper**

**Rating:** 6
**Confidence:** 3

**Review:**

The paper proposes an idea of jointly optimizing the sampling policy and predictor in NAS. With this method, we avoid to train a predictor which performs well on the whole search space and search a model with less queries.

### advantages
The authors propose that training a good accuracy predictor for models in the whole search space is difficult. Thus, the paper focuses on a predictor that works in a small range, then optimizes the sampling policy as well as the predictor to yield a good structure.

Compared with some evolution-based and RL-based methods, the algorithm observes a better model with less queries.

Totally speaking, the idea is very interesting and it does ease the search process.

### Weaknesses
A weakness of the paper is that it does not compare with differentiable methods such as DARTS. It is well known that differentiable methods are nearly the fastest methods now and yield not bad results.

As the results show, the algorithm finds a good model on ImageNet with 1000 queries, which seems to take more time than differentiable methods.

### Question
1. For each query, do we need to train the model till convergence?
2. How much time it takes to yield a good model (i.e., time for 1000 queries)?

---

> ### Author Response · Authors · 2020-11-23
> **Response to Reviewer2**
>
> *Q1: Comparison with DARTS*
>
> Sample-based NAS (our method) and gradient-based NAS (DARTS) are two different streams in the NAS field, the latter usually has worse performance and but is more efficient since they do not need to train each architecture from scratch. To give the reader a boarder context, we added the comparison with SoTA methods in Differentiable NAS[1][2][3][4] in the following Table 1. Additionally, we also cite the result of DSNAS[5]. As we can see, our method and sample-based NAS usually achieve higher performance than gradient-based NAS methods. Moreover, regarding the efficiency, since we only use the accuracy derived from weight-sharing supernet to train our NAS predictor, our weak NAS predictor has a comparable efficiency to gradient-based NAS.
>
> Table 1: Compared to SOTA results on ImageNet using NASNet search space.
>
> | Model | Methods|\#Queries|Top1 Err.(\%)|Top5 Err.(\%)|Params(M)|FLOPs(M)|GPU Days|
> |----|:----:|----:|----:|----:|----:|----:|----:|
> | MobileNetV2  | Manual | -  | 25.3 |  -  | 6.9 | 585 | - |
> | ShuffletNetV2 | Manual | - | 25.1 |  -  | 5.0 | 591 | - |
> | EfficientNet-B0 | Manual | - |   23.7 | 6.8 | 5.3 | 390 | - |
> | SNAS[1] | Gradient-based | - | 27.3 | 9.2 | 4.3 | 522 | 1.5 |
> | DARTS[2] | Gradient-based | - | 26.9 | 9.0 | 4.9 | 595 | 4.0 |
> | P-DARTS[3] | Gradient-based | - | 24.4 | 7.4 | 4.9 | 557 | 0.3 |
> | PC-DARTS[4] | Gradient-based | - | 24.2 | 7.3 | 5.3 | 597 | 3.8 |
> | DS-NAS[5] | Gradient-based | - | 24.2 | 7.3 | 5.3 | 597 | 10.4 |
> | NASNet-A | Sample-based | 20000 | 26.0 | 8.4 | 5.3 | 564 | 2000 |
> | AmoebaNet-A  | Sample-based | 10000 | 25.5 | 8.0 | 5.1 | 555 | 3150 |
> | PNAS         | Sample-based | 1160 | 25.8 | 8.1 | 5.1 | 588 | 200 |
> | NAO  | Sample-based | 1000 | 24.5 | 7.8 | 6.5 | 590 | 200 |
> | Ours | Sample-based + Weight Sharing| 1000 | **23.6** | **6.8** | **5.9** | 597 | 1.08 |
>
> *Q2: For each query, do we need to train the model till convergence?*
>
> In all our experiment, we need to train each model till convergence.
> However, we found that our method still works if each query is trained for fewer epochs. As we can see in the following Table 2, training for 180 epochs can achieve similar performance as a fully trained model (200 epoch), but if we further decrease the training to 150/160 epochs, it will hurt the final performance dramatically due to the bad ranking of early stopping models.
>
> Table 2: Ablation on Training Epoch on NASBench-201 over 50 runs (CIFAR100)
>
> | Method   |      \#Queries      |  Test Acc.(\%) |  Std(\%) |
> |----------|:-------------:|:------:|:------:|
> | Ours (150 Epochs) | 200 | 72.57 (-0.86) | 0.48 |
> | Ours (160 Epochs) | 200 | 72.34 (-1.09) | 0.42 |
> | Ours (170 Epochs) | 200 | 72.67 (-0.76) | 0.38 |
> | Ours (180 Epochs) | 200 | 73.35 (-0.08) | 0.15 |
> | Ours (190 Epochs) | 200 | 73.39 (-0.04) | 0.13 |
> | Ours (200 Epochs) (Fully-Trained) | 200 | 73.43 (Baseline) | 0.12 |
>
> *Q3: How much time it takes to yield a good model (i.e., time for 1000 queries)?*
>
> In the ImageNet experiment, we only use accuracies derived from the supernet. In particular, we only evaluate a sample (i.e., a sub-network in the supernet) on a small subset of the test set, which takes around 6-7s. In total, evaluating 1000 samples takes around 1.6 hrs.
>
> [1] Xie, Sirui, et al. "SNAS: stochastic neural architecture search." arXiv preprint arXiv:1812.09926 (2018).
> [2] Liu, Hanxiao, et al. "Darts: Differentiable architecture search." arXiv preprint arXiv:1806.09055 (2018).
> [3] Chen, Xin, et al. "Progressive differentiable architecture search: Bridging the depth gap between search and evaluation." Proceedings of the IEEE International Conference on Computer Vision. 2019.
> [4] Xu, Yuhui, et al. "Pc-darts: Partial channel connections for memory-efficient differentiable architecture search." arXiv preprint arXiv:1907.05737 (2019).
> [5] Hu, Shoukang, et al. "DSNAS: Direct Neural Architecture Search without Parameter Retraining." Proceedings of the IEEE/CVF Conference on Computer Vision and Pattern Recognition. 2020.

---

### Official Review · AnonReviewer4 · 2020-10-27
**This paper deals with a common bottleneck in neural search approches which is the prediction of performance of the potential sampled architectures. It proposes to learn weak predictors on a limited sample rather than model the performance over the whole architecture space. The experiments demonstrates that the proposed methods can reach a top performing architecture with fewer samples.**

**Rating:** 6
**Confidence:** 3

**Review:**

Overall this work moves into the right direction of trying to improve the performance of predictors.

Pros:

- The paper provides experiments on different datasets, and evaluates different predictors to validate the approach.

- The approach seems to have a significant speedup on the search time, but I would also like to see the results in terms of GPU days which is a common metric.

Cons:

- In a space where there is rapid progress and accuracy alone is not a practical metric for real-world applications since neural networks are used widely today from data centers to mobile devices. More effort should be towards this multi-objective problems considering also memory and computational cost. In this respect I find this a weakness of the paper in that it does not address these factors which would make the problem much more interesting, even though the proposed approach is appealing.

- A broader comparison should be made with DARTs approaches that have been shown to be generally faster. Also how does the approach compare with DSNAS: Direct Neural Architecture Search without Parameter Retraining in terms of efficiency?

- The setup and hardware used to evaluate the approach is not reported.

- Please correct for typos and grammatical mistakes, e.g., end of page 2 "the the loss", page 5 "our method can performs"

---

> ### Author Response · Authors · 2020-11-23
> **Response to Reviewer4**
>
> We appreciate your time and efforts in reviewing our paper!
>
> *Q1: Our method does not take into account of memory and computational cost constrains.*
>
> We can easily integrate above two constraints into the sampling stage. Instead of sampling from the whole search space, we only sample new architectures that meet specific memory and FLOPs requirements.
>
> *Q2: Comparison with Differentiable-NAS (Gradient-based NAS) and DSNAS in terms of efficiency.*
>
> It's not intuitive to directly compare non-differentiable (sample-based) NAS with differentiable (gradient-based) NAS since these are two different streams in the NAS literatures. The latter usually has worse performance but is more efficient since they do not need to train each architecture from scratch. To give the readers a boarder context, we added the comparison with SOTA differentiable NAS[1][2][3][4] in the following Table 5. In addition, we include the results of DSNAS [5]. As we can see in Table 5, our method and sample-based NAS achieve higher performance than gradient-based NAS methods. In terms of efficiency, since we only use the accuracy derived from the weight-sharing supernet to train our NAS predictor, our weak NAS predictor still has a comparable efficiency to gradient-based NAS.
>
> Table 1: Comparison results to more SOTA methods on ImageNet by using NASNet search space.
>
> | Model | Methods|\#Queries|Top-1 Err.(\%)|Top-5 Err.(\%)|Params(M)|FLOPs(M)|GPU Days|
> |----|:----:|----:|----:|----:|----:|----:|----:|
> | MobileNetV2  | Manual | -  | 25.3 |  -  | 6.9 | 585 | - |
> | ShuffletNetV2 | Manual | - | 25.1 |  -  | 5.0 | 591 | - |
> | EfficientNet-B0 | Manual | - | 23.7 | 6.8 | 5.3 | 390 | - |
> | SNAS[1] | Gradient-based | - | 27.3 | 9.2 | 4.3 | 522 | 1.5 |
> | DARTS[2] | Gradient-based | - | 26.9 | 9.0 | 4.9 | 595 | 4.0 |
> | P-DARTS[3] | Gradient-based | - | 24.4 | 7.4 | 4.9 | 557 | 0.3 |
> | PC-DARTS[4] | Gradient-based | - | 24.2 | 7.3 | 5.3 | 597 | 3.8 |
> | DS-NAS[5] | Gradient-based | - | 24.2 | 7.3 | 5.3 | 597 | 10.4 |
> | NASNet-A | Sample-based | 20000 | 26.0 | 8.4 | 5.3 | 564 | 2000 |
> | AmoebaNet-A  | Sample-based | 10000 | 25.5 | 8.0 | 5.1 | 555 | 3150 |
> | PNAS         | Sample-based | 1160 | 25.8 | 8.1 | 5.1 | 588 | 200 |
> | NAO  | Sample-based | 1000 | 24.5 | 7.8 | 6.5 | 590 | 200 |
> | Ours | Sample-based + Weight Sharing| 1000 | **23.6** | **6.8** | **5.9** | 597 | 1.08 |
>
> *Q3: The setup and hardware used to evaluate the approach is not reported*
>
> We conduct all experiment on Nvidia Tesla P100 GPUs with 16GB VRAM.
>
> *Q4: Typos and grammatical mistakes*
>
> Thanks for pointing out, we will correct them in the camera-ready manuscript.
>
> [1] Xie, Sirui, et al. "SNAS: stochastic neural architecture search." arXiv preprint arXiv:1812.09926 (2018).
> [2] Liu, Hanxiao, et al. "Darts: Differentiable architecture search." arXiv preprint arXiv:1806.09055 (2018).
> [3] Chen, Xin, et al. "Progressive differentiable architecture search: Bridging the depth gap between search and evaluation." Proceedings of the IEEE International Conference on Computer Vision. 2019.
> [4] Xu, Yuhui, et al. "Pc-darts: Partial channel connections for memory-efficient differentiable architecture search." arXiv preprint arXiv:1907.05737 (2019).
> [5] Hu, Shoukang, et al. "DSNAS: Direct Neural Architecture Search without Parameter Retraining." Proceedings of the IEEE/CVF Conference on Computer Vision and Pattern Recognition. 2020.

---

### Official Review · AnonReviewer1 · 2020-10-29
**Interesting approach**

**Rating:** 6
**Confidence:** 3

**Review:**

Summary of contribution: The authors propose an interesting approach to address the sample-efficiency issue in Neural Architecture Search (NAS). Compared to other existing predictor based methods, the approach distinguishes itself by progressive shrinking the search space. The paper correctly identifies the sampling is an important aspect in using a predictor based NAS method;

The writing is clear and easy to follow. The author provided an explanation of how their algorithm works, evaluate the algorithm on both NASBench-101 and NASBench-201, and show their methods work in practice on both

Some critiques:

1. Questions on predictor methods:

I understand the NAS community has accepted many predictor based paper in major conferences, but it seems NAS is re-making the wheel in derivative-free optimizations, e.g. Bayesian Optimization, Evolutionary Algorithm, and MCTS. The predictor is essentially the surrogate model in Bayesian Optimizations, however, the current predictor methods simply ignore the acquisition function in BO that makes the trade-off between exploration and exploitation. These predictor paper in NAS basically suggest the acquisition functions are not necessary, and we can achieve great performance without that (e.g. on NASBench, I will come to this later). Apparently this predictor trend in NAS community is against the decades of development of Bayesian Optimization. If the authors believe the development in BO community is wrong, please justify that using acquisition is indeed not important, i.e. exploration is not necessary, with extensive experiments.

If the new experiment can persuade me, I believe this paper can be top 1% paper, and I will argue for accepting it.

Here I'd like to list a set of predictor based paper that I have seen in the NAS community:

[1] Bridging the Gap between Sample-based and One-shot Neural Architecture Search with BONAS, NeurIPS-2020

[2] Brp-nas: Prediction-based nas using gcns, NeurIPS-2020

[3] Neural predictor for neural architecture search, ECCV-2020

I understand these paper have been accepted but do not necessarily mean these approaches expand the frontier of knowledge.

2. Evaluations on NASBench.

The reason why predictor works well on NASBench simply because it can predict every architectures. The largest dataset has 4.2*10^5 architectures, it won't be hard to predict them all. However, I believe the original NASBench paper has set baselines for us to compare (though currently published paper fail to follow these baselines and setup). I strongly encourage the authors to take look at the following repository, to see how original NASBench-101 setup the comparisons. I admit the design of NASBench might have some glitch in evaluations, but it is important to follow the same standard.
https://github.com/automl/nas_benchmarks

3. Evaluations on ImageNet.

Getting a good results on CIFAR10 and ImageNet can be tricky. Given the current ImageNet accuracy is 76.4@597 MFLOPS, while the SoTA top1 accuracy for a 600 MFLOPS model is 80.8. I understand these models use different search spaces, but the results on CIFAR-10 is missing. There are too many tricks to hack the network accuracy on ImageNet. Therefore, it will be better to judge NASNet search space on CIFAR-10.

---

> ### Author Response · Authors · 2020-11-23
> **Response to Reviewer1 (part2/2)**
>
> *Q2: Our method does not follow the same guideline used in NASBench-101?*
>
> We exactly follow the guideline described in the Section 2.6 and Supplement Section 6 of the NASBench-101 [4] paper
>
> - "Repeatedly query the dataset at $(A, E_{stop})$ pairs to train our NAS predictor, where A is an architecture in the search space and $E_{stop}$ is an allowed number of epochs ($E_{stop} \in \{4, 12, 36, 108\}$). Each query does a look-up using a random trial index, drawn uniformly at random from $\{1, 2, 3\}$, to simulate the stochasticity of SGD training."
>
> In our paper, we followed the exact setting. For each sample fed into our weak predictors, we queried a random trial index in the look-up using $E_{stop}$=108 epoch.
>
> - "Perform many runs of the various NAS algorithms."
>
> In Table 1/2/3 and Figure 4/5/6/7 of our paper, We performed 250 runs of various NAS algorithms and reported the mean and variance.
>
> - "Plot performance as a function of estimated wall clock time and/or number of function evaluations. This allows judging the performance of algorithms under different resource constraints."
>
> In Table 1/2/3/4 and Figure 4/5/6/7 of our paper, We plotted performance as a function of number of evaluations calls (number of samples).
>
> - "Do not use test set error during the architecture search process. In particular, the choice of the best architecture found up to each time step can only be based on the training and validation sets. The test error can only be used for offline evaluation once the search runs are complete."
>
> In our paper, we did not use test error until search is completed, details are described in Section 3.1.
>
> In summary, we think we have strictly followed the guideline in NASBench-101.
>
> *Q3: There are too many tricks to hack the accuracy on ImageNet, request evaluations on NASNet search space with CIFAR-10*
>
> In our ImageNet experiments, we followed standard Imagenet training [5], and **did not use any tricks** (e.g., MixUp, ZeroGamma, NoBiasDecay, DropPath, DropPath, EMA, etc). Based on the standard training, we still achieve SoTA performance on ImageNet NASNet search space with 23.6% Top-1 Error, shown in Table 4 in the paper. We will release the code upon acceptance.
>
> We further evaluate different methods on the CIFAR-10 dataset. Here, all methods including ours do not use Cutout for fair comparison, since Cutout is known as a useful trick to impact the performance on CIFAR-10. As shown in the Table 1, our method obtains the Top-1 Error of 3.42% at the cost of 4.3M Params, which is better than most of the SoTA methods. Although it is marginally worse than AmoebaNet-A and PNAS, our search time is **12,600X and 900X faster** than theirs respectively.
>
> Table 1. Compared to SOTA results on CIFAR-10 by using NASNet search space
>
> | Method         | \#Queries | Error (\%) |  # Params |  GPU Days |
> |----------      |:-----------:|------:|------:|------:|
> | Random-WS      | -    | 3.92 | 3.9M | 0.25
> | AmoebaNet-A    | 20000| 3.34 | 3.2M | 3150.00
> | PNAS           | -    | 3.41 | 3.2M | 225.00
> | ENAS           | -    | 3.54 | 4.6M | 0.45
> | NAO (Weight Sharing)         | -    | 3.53 | 2.5M | 0.30
> | Ours (Weight Sharing)        | -    | 3.42 | 4.3M | 0.25
>
> [4] "Nas-bench-101: Towards reproducible neural architecture search." ICML 2019.
> [5] https://github.com/pytorch/examples/tree/master/imagenet

---

> ### Author Response · Authors · 2020-11-23
> **Response to Reviewer1 (part1/2)**
>
> We appreciate your time and efforts in reviewing our paper!
>
> *Q1: Predictor-based NAS is re-making the wheel in derivative-free optimizations, such as Bayesian Optimization. The NAS predictor is essentially the surrogate model in Bayesian Optimizations, but it does not have an acquisition function thus lacks exploration of new architectures*
>
> * "Connection to Bayesian Optimization."
>
> Interpreting our Weak Predictor NAS as Bayesian Optimization (BO) is an interesting angle, but we respectfully suggest that might not represent the fully accurate picture. Please kindly allow us to explain:
>
> - Our current approach does not rely on any explicit uncertainty-based modeling such as Gaussian Process, mainly motivated by reducing the search cost (uncertainty modeling is heavier). However, our approach is not free of exploration – just that we have a different, simpler build-in exploration mechanism.
>
> - Specifically, at each iteration, instead of choosing all of Top K models by the predictor, we sample a fix amount N models in Top-K to explore new architectures in a probabilistic manner. Hence, an exploitation-exploration trade-off can be performed here, by adjusting the K to N ratio, and by varying the sampling strategy (follow uniform, linear or exponential distribution). Moreover, note that the weak predictor's fitting ability also gradually grows along with the search process, which implicitly introduces another source of possible "exploration" - and how well the predictor is fit can implicitly calibrate the balance between exploration and exploitation too. More details please see below.
>
> We find our simplified exploration in this current setting works well for this application, presumably due to the specially structured NAS search space in current use. Specifically:
>
> - Overall, NAS space has a highly non-uniform distribution of architectures, whose density is also heterogenous (although a precise definition of architecture distribution or density hinges on the concrete definition of distance). That is because the candidate architectures were created from varying operators (often causing similar performers), to varying width or depth (often causing bigger performance gaps). Therefore, the architectures display highly clustered distribution (as can be visualized by histogram on NAS-Bench-101/201 in Figure 8 of Supplementary); and the best performers are often gathered close to each other. This is also our work’s underlying assumption: we can progressively connect the line from the initialization to the finest subspace, where the best architecture resides.
>
> - Due to the non-unform or heterogenous assumption, the demand for the exploitation-exploration trade-off is also heterogenous across different search stages. At the beginning stage, more exploration will be needed to identify the promising direction towards the finer space. That is naturally achieved because the weak predictor at the initial stage only roughly fits the whole space (in other words, it CANNOT fit too well by then, and we intentionally make so). As we keep zooming-in to the good-performing subspace, the weak predictor is also gradually refined and better fit, therefore making stronger exploitation. Eventually, our weak NAS predictor is made to adaptively balance the exploration and exploitation throughout the search process, which echos the findings by many prior works such as [1][2][3].
>
> To summarize in brief: (1) our method does not take a precise BO form nor was it motivated that day, but it follows a similar spirit; (2) we use a different and simpler exploration mechanism, due to the specially structured NAS space; (3) we actually benefit from adaptively balancing exploration and exploitation along with search, in an implicit way.
>
> We are happy to include the above discussions into our revised paper. We sincerely appreciate the reviewer for his/her very insightful comments.
>
> [1] "AlphaX: eXploring Neural Architectures with Deep Neural Networks and Monte Carlo Tree Search." AAAI 2019.
> [2] "Sample-efficient neural architecture search by learning action space." arXiv 2019.
> [3] "Revisiting Neural Architecture Search." arXiv 2020.

---

### Decision · Program_Chairs · 2021-01-07
**Final Decision**

**Decision:**

Reject

**Comment:**

In line with recent work in the NAS literature, the authors consider a weak NAS performance strategy to filter out bad architectures and narrow down the exploration to the most promising region of the search space. The authors propose to estimate weak predictors progressively by learning a series of weak predictors that can connect towards the best architectures. The authors provided a number of additional experiments during rebuttal, addressing most of the reviewers' comments convincingly and further showing the strong performance of their method. However, the authors should relate their work to Bayesian optimization, which comes in many flavors, and black-box optimization techniques in general as their work shows a number of similarities, but is less principled.